# COVID-19 Vaccination Policies: Ethical Issues and Responsibility

**DOI:** 10.3390/vaccines10101602

**Published:** 2022-09-23

**Authors:** Maricla Marrone, Luigi Buongiorno, Alessandra Stellacci, Gerardo Cazzato, Pasquale Stefanizzi, Silvio Tafuri

**Affiliations:** 1Section of Legal Medicine, Department of Interdisciplinary Medicine, Bari Policlinico Hospital, University of Bari, 70124 Bari, Italy; 2Department of Emergency and Organ Transplantation, Section of Pathology, University of Bari “Aldo Moro”, 70124 Bari, Italy; 3Interdisciplinary Department of Medicine, Aldo Moro University of Bari, Piazza Giulio Cesare 11, 70124 Bari, Italy

**Keywords:** SARS-CoV-2, COVID-19, ethics, heard effect, responsibility

## Abstract

The World Health Organization (WHO) declared coronavirus 2 (SARS-CoV-2, COVID-19) a global pandemic on 11 March 2020. The emergence of the reliability of vaccines, the fear of possible vaccination-related side effects, and mass-media information created situations in which families and even health professionals developed hesitations regarding the need for vaccines, with a consequent decrease in vaccination coverage. This study discusses ethical issues and responsibility for the possible side effects of SARS-CoV-2 vaccination raised by vaccination policies.

## 1. Introduction

The World Health Organization (WHO) declared coronavirus 2 (SARS-CoV-2, COVID-19) infection a global pandemic on March 11, 2020 [1]. According to the WHO, over 512 million confirmed SARS-CoV-2 infections and over 6.2 million deaths in 223 countries were reported as of 5 May 2022. In Europe, the significant efficacy and safety of the four licensed vaccines—mRNA-1273 (Moderna), Ad26.COV2. S (Johnson & Johnson-Janssen), Chadox-1 (AstraZeneca), and BNT162b2 (Pfizer-BioNTech)—against severe acute respiratory syndrome from coronavirus 2 (SARS-CoV-2) have allowed the granting of authorization for the use of an emergency drug (vaccine) [2,3,4]. Vaccines for COVID-19 were welcomed with enthusiasm, and as of 5 May 2022, 4,634,230,468 people worldwide (58.1% of the population) have been fully vaccinated (https://covid19.who.int (accessed on 5 May 2022). Despite the efforts to ensure the distribution of vaccines to produce individual and collective benefits and the high adherence to the anti-SARS-CoV-2 vaccination offer, at the time of writing this manuscript, the WHO continues to identify vaccine hesitation as one of the top ten global health threats [5]. In this regard, the “pro-vax” and “anti-vax” debates have accompanied vaccinations’ history since the introduction of the smallpox vaccine at the end of the eighteenth century, with doubts arising about the efficacy and necessity of vaccines and the possibility of damage caused by them [6,7]. This debate was reinvigorated in the 1970s when several abortions occurred following the administration of vaccines developed from cell lines derived from aborted fetuses intended for research [8,9,10,11,12,13,14,15]. The vaccines produced by Oxford, AstraZeneca, and Johnson & Johnson were developed to target the SARS-CoV-2 infection, using cells replicated from HEK-293 cell lines obtained from fetuses after elective abortions to produce parts of the spike protein [9].

This abortive theme was of concern to some religious associations who focused on producing vaccine procedures [16].

Moreover, the most redundant mass-media campaigns deployed against vaccination led to some subjects’ uncertainty and doubts about the need for vaccination, with a consequent decrease in vaccination coverage, contributing to SARS-CoV-2 diffusion [17,18,19,20,21].

Other religious communities prioritized the achievement of herd immunity, identifying a spiritual, moral, and social obligation in vaccination [13,22].

With regard to the no-vax ideology, Burgess et al. distinguished “*between people wholly opposed to vaccination (anti-vaxxers) and individuals with limited or inaccurate health information or who have genuine concerns and questions about any given vaccine, its safety, and the extent to which it is being deployed in their interests before accepting it (vaccine hesitancy).*”.

A large-scale retrospective study published in 2020 in “*The Lancet*” has shown that vaccine confidence in their importance, safety, and effectiveness has improved in several countries worldwide [10].

However, due to the multifactorial elements, data from the World Health Organization (WHO) showed that 35.5% of the world population had not yet received any anti-SARS-CoV-2 vaccine on 5 May 2022.

One of the objectives of the WHO is to achieve herd immunity, which is principally influenced both by the efficacy of the vaccine and by the adherence of the population to vaccination plans [17].

The herd immunity concept can be described in the following terms: “*If a sufficient proportion of the population is immune—above the ‘herd immunity threshold’—then transmission generally cannot be sustained*”. However, it is often misconceived and is critical to long-term disease control [18].

With regard to the WHO’s aim, several authors questioned the lawfulness of achieving herd immunity through mandatory vaccination, mainly the modality of how the European Union approved the distribution of the SARS-CoV-2 vaccine [19].

Kowalik et al. highlighted how favoring the attainment of herd immunity gave priority to profit rather than to the safety of citizens, arguing that there is not a solid ethical basis for discriminating against the unvaccinated, demonstrated by the following statement: “*If I must accept an increased risk to myself in order to reduce the risk to others because everyone has a moral obligation to do so, then justice demands that others must also accept an increased risk to themselves in order to reduce the risk to me, therefore,*
*a*
*contradiction*.” [9].

Despite numerous conflicting opinions, vaccination plans have not yet been widely debated regarding mandatory vaccination, responsibility, and marketing authorization.

This study aims to discuss ethical issues and responsibility for the possible side effects of SARS-CoV-2 vaccination raised by vaccination policies.

This document contains no value judgment on the efficacy of the vaccine against COVID-19, nor does it seek to discredit the role played by vaccine manufacturers. It focuses only on assessing the ethical and protective implications for citizens. The aim is to fuel the debate on the conditions and effects of public health measures in a pandemic context.

## 2. How Are New Drugs Approved by the European Union?

The race toward a vaccine against COVID-19 has profoundly marked 2020 and 2021, during which a global planetary effort was dedicated to the research, production, authorization, and distribution of vaccines. Europe has taken a leading role in ensuring the availability of vaccines to European citizens and facilitating their distribution.

The European Parliament and the Council of the European Union (EU) with the regulation no. 726/2004 of 31 March 2004 have regularized and harmonized the procedures for the approval and supervision of the drug. Before a drug, in this specific case a vaccine, can be approved in the EU, it must undergo rigorous testing and, therefore, careful scientific evaluation by the regulatory authorities. These include the European Medicines Agency (EMA) and other regulatory bodies of individual EU (European Union) and EEA (European Economic Area) countries.

In the interest of public health, authorization decisions under the centralized procedure must be taken based on objective scientific criteria of the quality, safety, and efficacy of the medicinal product in question, excluding economic and other considerations.

However, in response to public health threats, such as the current pandemic, the EU has a specific regulatory tool to allow for the rapid availability of medicines for use in an emergency.

In such situations, the conditional marketing authorization (CMA) procedure is designed to allow authorization as quickly as possible as soon as sufficient data are available.

The critical issue is the balance between the need to speed up the authorization process to rapidly distribute vaccines and the obligation to ensure their quality, safety, and efficacy under EU law.

The CMA procedure provides the EU with a robust framework for rapid approval, security, safeguards, and post-authorization checks.

In general, before a vaccine can be approved in the EU, it must undergo rigorous testing and scientific evaluation by regulatory authorities.

As part of the trials, the vaccine quality is initially checked (Phase 1: drug discovery) regarding its composition (including excipients) and the production method.

Subsequently, there is a phase of verification of the effects of the vaccine through laboratory and animal experiments (Phase 2: pre-clinical), which, if successful, is followed by a clinical trial program on humans.

Clinical trials (Phase 3: clinical), in turn, include three phases, each of which is attended by a progressively increasing number of people.

This program must comply with strict rules and the procedures and protocols established by the regulatory authorities. At the end of the trial program, the manufacturer must apply for marketing authorization for the vaccine.

The application is submitted to the EMA, which assesses the vaccine’s safety, efficacy, and quality. If the EMA makes a recommendation, the Commission can proceed to authorize the marketing of the vaccine on the EU market [23].

According to the available data, the vaccine development procedure typically takes five to fifteen years [24].

The modality used to speed up the approval process is “cyclic reviews” (rolling reviews), which allowed the EMA to start evaluating vaccine data rather than waiting for the end of all trial phases.

The cyclical review evaluates the data on the quality of the vaccine and the results of the laboratory studies, considerably reducing the regular evaluation times, while continuing to guarantee the principles of quality, safety, and efficacy.

This way, when marketing authorization is applied, the formal assessment can proceed much faster. The data have already been analyzed as part of the cyclical review to issue conditional marketing authorization (CMA).

In addition to certifying that the safety, efficacy, and quality of the vaccine are proven and that the benefits of the vaccine outweigh the risks, it allows developers to submit additional data on the vaccine, even after the marketing authorization (contrary to standard authorizations, for which all data must be submitted prior to release).

The European Commission also ensures that the procedure for marketing authorization can be carried out as quickly as possible, shortening the administrative process.

On the other hand, the CMA can be converted into standard marketing authorization (SMA), as the marketing authorization holder fulfills its obligations and the complete data confirm that the medicine’s benefits continue to outweigh its risks (Figure 1).

If new data show that the medicine’s benefits no longer outweigh the risks, EMA may suspend or withdraw the marketing authorization [25].

Therefore, the choice of response results from a discretionary decision based on the level of risk deemed acceptable by the authorities. However, the discretion of decision makers is not unlimited, as they must comply with a series of procedural obligations to ensure that, even in conditions of scientific uncertainty, the use of the precautionary principle is supported by scientific data and is not arbitrary.

## 3. Safeguards and Responsibilities

Vaccines also have unpredictable and not wholly preventable side effects, as is the case with any drug on the market.

Specifically, the vaccines against SARS-CoV-2 have raised considerable doubts from a scientific point of view about their quality, effectiveness, and short and long-term risks.

The mass media have repeatedly ridden this wave, relying on the uncertainties of the population and discouraging vaccination against SARS-CoV-2.

Special legislation protects against vaccine damage in Italy (Law 210/1992). However, beyond the statutory social security protection, the request for compensation for vaccine damage is often formulated in civil proceedings. In terms of liability, to obtain compensation for the damage, it would be necessary to demonstrate that the patient has undergone vaccination and that there has been consequential damage resulting from the treatment intended as an unforeseeable or unavoidable complication [26,27].

Therefore, many of the symptoms reported by vaccinated subjects fall into mild and temporary effects.

On this point, it should be specified that vaccines are included in the product as established by Directive 374/85 EEC, which relates to the Member States’ laws, regulations, and administrative provisions, regarding liability for damage from defective products.

The question the authors ask is who is responsible if a subject suffers permanent damage or dies after vaccination?

Is the pharmaceutical company that produced the vaccine responsible?

Can a drug that has undergone the necessary testing phases and passed the checks be considered “defective”, after obtaining marketing authorization from the competent authorities, as already illustrated in the dedicated section of the following manuscript?

We believe it is impossible to eliminate the destructive potential that is intrinsic to each medicine, nor is it conceivable to prohibit its use if it means losing all the resulting benefits. In this sense, as mentioned, the European Union recognizes the risk as “acceptable” when there is a correct balance between risks and benefits with the use of the product.

Is the country that authorizes the marketing/administration of the vaccine responsible?

According to Article 12 of the International Covenant on Economic, Social, and Cultural Rights (ICESCR), Member States have an obligation to prevent and control epidemics by guaranteeing access to essential medicines for all [28].

In Italy, in 1992 and 2005, an indemnity system was introduced, which provides the disbursement by the Ministry of Health of certain subsidies in favor of people who have suffered injuries or infirmities because of compulsory vaccination.

Ultimately, the Ministry of Health has an obligation to supervise and control public health. In this sense, one cannot deny the hypothetical responsibility of the Ministry, which, in violation of the surveillance and control obligations for the protection of public health, has not prohibited the use of a “dangerous” vaccine. Furthermore, considering that vaccination must not be understood as an individual preventive measure but as a public health action, it is legitimate for the injured party, with regard to an action carried out in the public interest, to receive adequate compensation from the community.

However, was the SARS-CoV-2 vaccine recommended?

In the case of the vaccine against SARS-CoV-2, the vaccination obligation for the whole population has not been established, but only for some subgroups (health care, people over the age of fifty). However, an organic provision was adopted by the Ministry of Health, which obliged the regions to offer active and free vaccination to the population (Decree of the Minister of Health, 2 January 2021), and legislative measures were adopted to limit freedom. For example, unvaccinated subjects were obliged to show the Green Pass for access to various social activities. Furthermore, in the presence of widespread and repeated communication campaigns in favor of vaccination, it is natural that a climate of trust on the part of the patient develops toward what is recommended for safeguarding the collective interest.

Are the health workers who vaccinate responsible?

Another responsibility hypothesis could be ascribed to the vaccinator, due to the lack or incorrect acquisition of informed consent.

From this point of view, however, it appears unlikely and unlikely that the patient, if informed of a risk of death of less than 0.05% following the inoculation and of a significantly higher risk of dying because of the infection of the virus, refuses to undergo vaccination.

Moreover, the issue of the “risk” of adverse events related to inoculation has been widely circulated in the media, especially for the vaccine produced by Astra Zeneca, making any thesis about a lack of knowledge of it seem unrealistic.

Another responsibility could be attributable to the health worker who carries out the administration of the vaccine, that is, the one who physically carries out the muscle injection.

However, this hypothesis appears challenging to configure since it is a routine practice and, therefore, not difficult to carry out.

Similar considerations could derive from other aspects related to the dosage or even the observance of the prescriptions contained in the indications provided when the vaccine was placed on the market.

As a rule, the vials delivered by the manufacturer contain multiple doses of the vaccine that must be reconstituted through a standard procedure and divided for every single injection, and it would be difficult to deny the fault of the doctor or vaccinator nurse who administered an incorrect amount of the drug.

However, this wrong gesture should be looked alongside other factors, such as tension, emergency, poor organization, agitating work rhythms, and, at least in the initial stages, may be characterized by a shortage of health personnel.

In this regard, in Italy, to cope with this situation and to protect health professionals, the law decree 44/2021 has provided the so-called criminal shield, guaranteeing that the health personnel who administer the vaccine will not receive punishment for the crimes of manslaughter and negligent personal injury when the administration is carried out, in compliance with the indications contained in the authorization provisions [29].

Another possibility is administering the drug, despite specific contraindications reported in the documentation relating to the marketing authorization that are not considered, or otherwise ignored, by the doctor.

In this case, while contextualizing the medical act in an emergency context, we believe a responsibility profile may emerge for the vaccinator.

However, responsibility cannot be recognized for those cases in which there were no contraindications to administration or were somewhat not reasonably foreseeable, since the fact that medicine involves “risks” is not a circumstance alone capable of generating liability and also alternatively, this does not mean that the drugs should no longer be used.

## 4. Bioethics and Vaccination Obligation

The balance between individual actions and their impact on collective health, as well as the assessment of the resulting risks and benefits, can generate ethical and ontological conflict [9].

The model of the four principles of bioethics formulated by Tom Beauchamp and James Childress in their text “*Principles of Biomedical Ethics*”, whose first edition dates to 1979 [30], recognizes four moral principles that must be used as a basis for judging bioethical problems to be contextualized to the single case of interest.

The four principles are as follows:-Principle of autonomy: the patient has the right to refuse treatment and participate in decision-making.-Principle of charity: healthcare personnel must act while protecting the patient’s interest.-Principle of non-maleficence or primum non-nocere: the healthcare staff must not cause harm to the patient.-Principle of justice: in case of limited resources, treatments must be distributed to patients fairly and reasonably.

Specifically, the principle of autonomy illustrates how autonomy is understood as the will of a person, which is exercised through informed consent or informed refusal [31,32] and that it must meet the following four mandatory criteria [33,34,35] for its legitimacy:-Decision-making capacity of a patient.-Adequate disclosure of information.-Adequate understanding of the information.-Voluntariness.

The difference between animals, inanimate objects, and humans is that the latter can freely decide between one course of action and another by appealing to the principle of free will. The nature of free will has been debated for centuries, and it is still challenging to obtain a clear definition of what is meant by it.

The concept of free will has a mostly philosophical history but also a partly theological one, and it was first conceived as a distinct power by Augustine [36].

For example, when a subject decides how to act and acts as he decides, he responds to situations with intelligence and integrity.

However, another subject, or the same subject in a different mood, might respond to the same situation very differently. This supports the fact that causal laws govern a person’s decision.

Specifically, according to the philosopher, our will is always “directed” to specific love objects, which can include external elements (food, sex, wealth) or internal elements (pride, goodness, truth) [37].

The theme of will has also been widely discussed in classical literature and Dante Alighieri identifies will as one of the two fundamental powers of the rational soul and the intellect, to which it is sometimes opposed [37].

In summary, free will is understood as the ability to choose to have a course of action to satisfy a desire without extrinsic or intrinsic constraints that dictate the choice [36,37]

However, human actions are conditioned by emotions and, in particular, by fear [26,36].

Fear is a reaction rooted in living beings that has the purpose of survival and protection from external threats [36].

This emotive reaction plays such an essential role in human/animal decision-making and also can take elementary forms, for example, the automatic retraction of a snail’s antenna, or can be very articulated, as in humans, where fear triggers mechanisms that start from the brain in response to a stimulus and involve the whole organism [26,38,39].

Whenever we are faced with a stimulus that is interpreted by our brain as a threat, various chain mechanisms are activated with a consequent release of stress hormones and activation of the sympathetic nervous system, which is involved in those functions defined as “*attack or leak*” [26].

Because of these conditionings in the subjects in the decision-making phase, fear by some authors is opposed to knowledge or to a choice free from conditionings [38,39].

Many counter-cultural currents related to vaccination policies leverage fear, presenting scary scenarios to citizens concerning the vaccine.

Several studies state that the spread of SARS-CoV-2 infection and poor adherence to vaccination campaigns is disproportionately more widespread among minorities, low-income people, and those with lower educational qualifications [18,19].

An unconditional choice, on the other hand, should be guided by a more lucid and coherent vision, following the principles of autonomy and bioethics.

In the case of vaccination against SARS-CoV-2, in the opinion of Rus et Groselj, the conditions envisaged by the four bioethical principles set out are primarily satisfied, including the principle of decision-making autonomy [40].

Regarding the COVID-19 vaccination, the Commission’s risk management before granting a CMA has raised many concerns, as it is based on preliminary data.

However, there is numerous scientific evidence that subsequently emerged to support the efficacy of vaccination against SARS-CoV-2.

A recent report, published by the Italian Ministry of Health on 27 April 2022, showed the terms of efficacy of vaccinations, highlighting how in Italy, the death rate for problems related to coronavirus infection was nine times higher in unvaccinated subjects than in those vaccinated [41].

Considering this, many authors have, in turn, published various advanced hypotheses and sometimes creative proposals in order to encourage individual citizens to get vaccinated. Navin and Largent proposed complicating the procedure for refusing vaccination [42].

Giubilini, on the other hand, appealing to the ethical principle of the “*less restrictive alternative*” proposed incentive formulas centered on push policies, monetary incentives, penalties for non-vaccination, and withdrawal of state benefits for families who do not vaccinate their children [43].

In addition, Flanigan, to sensitize unvaccinated subjects, metaphorically compared the subject who chooses not to be vaccinated to a subject who shoots a gun in the air, endangering the lives of innocent bystanders [44].

Jason Brennan instead argued that we should stick to the “*clean hands principle*”, according to which there is a moral obligation of the individual not to participate in collectively harmful activities, even if the outcome is overdetermined [45].

To encourage vaccination, Brennan et al. used the example of a firing squad in which a gang of ten snipers is about to kill an innocent child. The question that Brennan poses to the reader is whether, despite the impossibility of preventing the child’s killing, he is willing to join the firing squad, meaning the innocent child is the fragile subject to be protected from SARS-CoV-2 [45].

The authors believe that all citizens have a collective moral obligation to aspire to achieve herd immunity to protect fragile individuals from eradicating the pandemic event in progress [11,15].

In line with this concept, the various campaigns in favor of vaccines should start with the awareness of the counter-cultural phenomenon and the consequent vaccination hesitation.

However, as already proposed by other authors, we believe that to address hesitation and resistance to vaccination, a multidisciplinary approach that involves social and behavioral change communication specialists, social marketers, medical anthropologists, psychologists, and practitioners is likely to be required [46].

We believe doctors are the figure for vaccination strategies that can strengthen vaccine confidence and reduce rejection rates. Local doctors, in particular, play a fundamental role in public health, not only in terms of treatment but, above all, in terms of prevention. Regarding the influence of the physician on the citizens, a study conducted in the U.S.A. showed that trust in experts and doctors and the fear of contracting COVID-19 were associated with a reduced risk of hesitation [20].

The authors believe that it is necessary to illustrate the importance and necessity of vaccination to the citizen, underlining how much vaccines have contributed to eradicating and controlling deadly diseases, whose concern today does not affect us precisely due to the success and effectiveness of vaccination [17]. The need to show the results obtained with the current vaccination and the related side effects using the mass media is also important.

Institutions should make vaccination against SARS-CoV-2 infection mandatory based on higher interest, invoking the primacy of the principle of social solidarity.

In the absence of these safeguards, the individual would be forced to bear any negative consequences that derive from the health treatment carried out in the interest of the community.

## 5. Conclusions

In conclusion, it can be said that the acceptance or rejection of the vaccine is the result of a decision-making process influenced by the confluence of different socio-cultural, political, religious, and personal factors [14].

The authors believe that the achievement of herd immunity, understood as a common good, should justify the momentary sacrifice of the individual’s right to self-determination on the altar of the pre-eminent health protection needs of the entire community through the introduction of a motivated mandatory vaccination, presided over by the indemnification of the damage resulting from the administration of the vaccine.

It is, therefore, essential that the interaction between the individual, collective and institutional responsibilities regarding vaccination decisions and policies remain at the center of future philosophical, sociological, and legal work on vaccination.

The authors hope that this contribution can offer food for thought on the public health measures adopted in conditions of risk and uncertainty.

## Figures and Tables

**Figure 1 vaccines-10-01602-f001:**
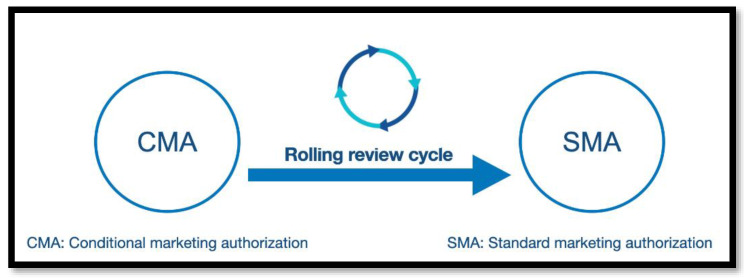
The conversion process from conditional marketing authorization to standard marketing authorization.

## Data Availability

Not applicable.

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
