# Peer review of "COVID-19 Vaccination Policies: Ethical Issues and Responsibility"

_vaccines, 2022, doi:10.3390/vaccines10101602_

Round 1
Reviewer 1 Report
The topics discussed in the paper are important and may be of interest for theaverage reader of the journal. However, there are limits and weaknesses that need to be addressed. First of all, the aim of the review/opinion paper. Do the authors want to address the problem of vaccine refusal/hesitancy, or ethical and/or legislative (i.e. mandates) issues. This should be clarified. For this reason, I find the title also misleading. As far as I understand, they want to discuss all these issue in the light of "conditional marketing authorization" of COVID-19 vaccines. Thus I would limit the discussion to ethical problems raised by vaccination policies such as, for example, mandatory vaccination (or just recommendations).
Specific points: the Introduction is long and rather confusing. Firstly they discuss the problems raised by vectored vaccines then they jump to ethical-religious issuesand finally to fear of vaccination. Please be more specific.
in page 2 row 84, WHO concerns about 35.5% of the population that has not been vaccinated should be interpreted with caution since it is not necessarily a consequence of lack of access.
The issue of "herd immunity" does not appear to be on the table any more. Thus better to clarify it (the consequence is that even vaccinating 95% of the population could not stop virus circulation.
Please do not use the term vaccine efficiency (but vbaccine efficacy). Again, do not use the term "reservoir" of the most intense viral circulation.
In page 4/5, the list of side effect is not useful .
On page 5, when discussiong about 'errors', these must be distinguished from other general responsibilities. Moreover, I don't think that the role of the MoH (that launches recommendations) is more important, from the point of view of the responsability, of that of regulatory agencies (on the basis of their resolutions, usually the MoH provides recommendations). This should be better discussed and clarified.
The conclusions and the take home message should be better emphasized.
Author Response
Reviewer #1:
Reviewer’s comment: “The topics discussed in the paper are important and may be of interest for the average reader of the journal. However, there are limits and weaknesses that need to be addressed. First of all, the aim of the review/opinion paper. Do the authors want to address the problem of vaccine refusal/hesitancy, or ethical and/or legislative (i.e. mandates) issues. This should be clarified. For this reason, I find the title also misleading. As far as I understand, they want to discuss all these issue in the light of "conditional marketing authorization" of COVID-19 vaccines. Thus I would limit the discussion to ethical problems raised by vaccination policies such as, for example, mandatory vaccination (or just recommendations).”
Authors ‘comment: We are grateful to the reviewer for her/his positive and encouraging comments. We found this comment very stimulating to continue deepening this field. As proposed by the reviewer, we change the title as follows: “Covid-19 vaccination policies: ethical issues and responsibility”. As suggested by the reviewer, we added the following sentence in the introduction section: “Despite numerous conflicting opinions, vaccination plans have not yet been widely debated regarding mandatory vaccination, responsibility, and marketing authorization. This study aims to discuss ethical issues and responsibility for the possible side effects of Sars-CoV-2 vaccination raised by vaccination policies.”
Reviewer’s comment: “Specific points: the Introduction is long and rather confusing. Firstly they discuss the problems raised by vectored vaccines, then they jump to ethical-religious issues, and finally to fear of vaccination. Please be more specific.”
Authors’comment: Thank the reviewer for the valuable advice of the reviewer; we streamlined the introductory section by simplifying some sentences to make it easier to read.
Reviewer’s comment: “in page 2 row 84, WHO concerns about 35.5% of the population that has not been vaccinated should be interpreted with caution since it is not necessarily a consequence of lack of access.”
Authors’comment: We are grateful to the reviewer for displaying a confusing element for readers that we have modified as follows: “A large-scale retrospective study published in 2020 in "The Lancet" has shown vaccine confidence in the importance, safety, and effectiveness has improved in several countries worldwide. However, due to the multifactorial elements, data from the World Health Organization (WHO) showed that 35.5% of the world population had not yet received any an-ti-Sars-coV-2 vaccine on 5 May 2022.”
Reviewer’s comment: “The issue of "herd immunity" does not appear to be on the table any more. Thus better to clarify it (the consequence is that even vaccinating 95% of the population could not stop virus circulation”
Authors’comment: Thank you for pointing this delicate shortcoming out. We have filled in as follows: “However, the Herd immunity concept that intends "If a sufficient proportion of the population is immune - above the 'herd immunity threshold' - then transmission generally cannot be sustained" is often misconceived and critical to long-term disease control.”
Reviewer’s comment: “Please do not use the term vaccine efficiency (but vbaccine efficacy). Again, do not use the term "reservoir" of the most intense viral circulation.”
Authors’comment: We thank the reviewer for the indication; we have replaced all the indicated terms in the paper.
Reviewer’s comment: “In page 4/5, the list of side effect is not useful”
Authors ‘comment: We thank the reviewer for the comment. We have removed the list of side effects.
Reviewer’s comment: “On page 5, when discussiong about 'errors', these must be distinguished from other general responsibilities. Moreover, I don't think that the role of the MoH (that launches recommendations) is more important, from the point of view of the responsability, of that of regulatory agencies (on the basis of their resolutions, usually the MoH provides recommendations). This should be better discussed and clarified.”
Authors’comment: We thank the reviewer for the comment, and we modified the section indicated by removing the references to evaluation errors as a possible source of misunderstanding for readers.
Reviewer’s comment: “The conclusions and the take home message should be better emphasized”
Authors’comment: Thank you for your suggestion. We have modified the discussion by emphasizing the contribution of our study to literature.
Reviewer 2 Report
In this manuscript, Maricla Marrone et al. presented a very interesting essay discussing some of the most important issues around SARS-CoV-2 vaccine development and application in society.
Throughout the manuscript, the authors touched upon a few issues surrounding SARS-CoV-2 vaccines, for example, the debate on the usage of HEK293 cells, a cell line derived from fetus after abortion, in the production of some SARS-CoV-2 vaccines; conditional marketing authorization versus standard marketing authorization; the responsibility following adverse effects after vaccination; free will and fear in decision making on individual and community levels; and the strategies to combat vaccine hesitance and to achieve better vaccine uptake hence herd immunity.
The authors wrote vaccination as the main strategies to achieve herd immunity in society. However, as we all have experienced multiple waves of SARS-CoV-2 omicron variants, which in general cause milder disease compared to earlier variants, wide spreading in communities, and repeated breakthrough infections from individuals fully vaccinated in 2022. The authors are invited to discuss the impact of these breakthrough infections on the vaccine hesitance and herd immunity in society.
The authors wrote around quite a few interesting issues. However, with most of the manuscript being written in short and single sentence paragraphs, it is difficult to get a clear structure and comprehensive understanding of the manuscript. In addition, the title of the manuscript does not well reflect what is discussed in the manuscript. The authors are invited to write in a more concise and well structure manner to help readers better understand the story.
The authors used “in Europe” in title. However, in the introduction, the usage of HEK293 cells is hot debated in Canada and the US, whilst in later sections 3 and 4 cases in Italy were used as examples when mentioning vaccine safeguards and vaccination obligation. More examples from other European countries would help broaden the scope.
Minor issues:
1. The abbreviation of coronavirus 2 is mentioned multiple times across the manuscript but with different variants, such as Sars-CoV2 in line 13; SARS-VOC-2 in line 42; SARS-COV-2 in line 86; and SARS-CoV-2 in line 92. Is it possible to use one version throughout the manuscript? The same issue applies to the abbreviation Covid 19 versus COVID-19.
2. “EEA” in line 122 needs explanation.
3. Some of the sentences are quite ambiguous and difficult to understand like lines 216-219. Other places it reads odd, such as “it appears unlikely and unlikely” in line 248 and “must be distributed to patients fairly and fairly” in line 299-300.
Author Response
Reviewer #2:
Reviewer’s comment: “In this manuscript, Maricla Marrone et al. presented a very interesting essay discussing some of the most important issues around SARS-CoV-2 vaccine development and application in society. Throughout the manuscript, the authors touched upon a few issues surrounding SARS-CoV-2 vaccines, for example, the debate on the usage of HEK293 cells, a cell line derived from fetus after abortion, in the production of some SARS-CoV-2 vaccines; conditional marketing authorization versus standard marketing authorization; the responsibility following adverse effects after vaccination; free will and fear in decision making on individual and community levels; and the strategies to combat vaccine hesitance and to achieve better vaccine uptake hence herd immunity.”
Authors’ comment: We are grateful to the reviewer for her/his positive and encouraging comments. As recommended, we have answered each of your points below.
Reviewer’s comment: “The authors wrote vaccination as the main strategies to achieve herd immunity in society. However, as we all have experienced multiple waves of SARS-CoV-2 omicron variants, which in general cause milder disease compared to earlier variants, wide spreading in communities, and repeated breakthrough infections from individuals fully vaccinated in 2022. The authors are invited to discuss the impact of these breakthrough infections on the vaccine hesitance and herd immunity in society.”
Authors’ comment: Thank you for your suggestion. We appreciate the indication given to us, but the article does not focus on the effectiveness or otherwise of herd immunity. For this reason, we have limited ourselves to highlighting the controversies that this concept may raise in the scientific community as follows:
“One of the objectives of the WHO is to achieve herd immunity, which is principally influenced both by the efficacy of the vaccine and by the adherence of the population to vaccination plans. However, the Herd immunity concept that intends "If a sufficient proportion of the population is immune - above the 'herd immunity threshold' - then transmission generally cannot be sustained" is often misconceived and critical to long-term disease control.”
Reviewer’s comment: “The authors wrote around quite a few interesting issues. However, with most of the manuscript being written in short and single sentence paragraphs, it is difficult to get a clear structure and comprehensive understanding of the manuscript. In addition, the title of the manuscript does not well reflect what is discussed in the manuscript. The authors are invited to write in a more concise and well structure manner to help readers better understand the story.
The authors used “in Europe” in title. However, in the introduction, the usage of HEK293 cells is hot debated in Canada and the US, whilst in later sections 3 and 4 cases in Italy were used as examples when mentioning vaccine safeguards and vaccination obligation. More examples from other European countries would help broaden the scope.”
Authors’ comment: Thank you for your suggestion. We have changed the title as suggested by the reviewer to specify better the topics covered and removed the term Europe. All the paper has been revised to make it more accessible for reading.
Reviewer’s comment: “The abbreviation of coronavirus 2 is mentioned multiple times across the manuscript but with different variants, such as Sars-CoV2 in line 13; SARS-VOC-2 in line 42; SARS-COV-2 in line 86; and SARS-CoV-2 in line 92. Is it possible to use one version throughout the manuscript? The same issue applies to the abbreviation Covid 19 versus COVID-19.”
Authors’ comment: Thank you for the helpful comment. We have unified in using the terms Covid-19 and Sars-CoV-2.
Reviewer’s comment: “EEA in line 122 needs explanation.”
Authors’ comment: Thank you for the valuable comment. We added a more precise explanation as follows:
“These include the European Medicines Agency (EMA) and other regulatory bodies of individual EU (European Union) and EEA (European Economic Area) countries.”
Reviewer’s comment: “Some of the sentences are quite ambiguous and difficult to understand like lines 216-219. Other places it reads odd, such as “it appears unlikely and unlikely” in line 248 and “must be distributed to patients fairly and fairly” in line 299-300.”
Authors’ comment: Thank you for your suggestion. We have made an overall correction of the article by resolving the typos you reported.
Round 2
Reviewer 1 Report
The MS is much improved. I have no further comments.